# Immunity onset alters plant chromatin and utilizes *EDA16* to regulate oxidative homeostasis

**Alonso J. Pardal**[1¤], **Sophie J. M. Piquerez**[1,2☙], **Ana Dominguez-Ferreras**[1☙], **Lucas Frungillo**[3], **Emmanouil Mastorakis**[1], **Emma Reilly**[1], **David Latrasse**[2], **Lorenzo Concia**[2], **Selena Gimenez-Ibanez**[4], **Steven H. Spoel**[3], **Moussa Benhamed**[2]*, **Vardis Ntoukakis**[1]*

**1** School of Life Sciences, University of Warwick, Coventry, United Kingdom, **2** Institute of Plant Sciences Paris-Saclay (IPS2), CNRS, INRAE, Université de Paris, Orsay, France, **3** Institute of Molecular Plant Sciences, School of Biological Sciences, University of Edinburgh, Edinburgh, United Kingdom, **4** Plant Molecular Genetics Department, Centro Nacional de Biotecnología-CSIC (CNB-CSIC), Madrid, Spain

☙ These authors contributed equally to this work.
¤ Current address: Division of Biomedical Sciences, Warwick Medical School, Coventry, United Kingdom
* moussa.benhamed@u-psud.fr (MB); V.Ntoukakis@warwick.ac.uk (VN)

**Data Availability Statement:** All relevant data are within the manuscript and its Supporting Information files.

## Abstract

Perception of microbes by plants leads to dynamic reprogramming of the transcriptome, which is essential for plant health. The appropriate amplitude of this transcriptional response can be regulated at multiple levels, including chromatin. However, the mechanisms underlying the interplay between chromatin remodeling and transcription dynamics upon activation of plant immunity remain poorly understood. Here, we present evidence that activation of plant immunity by bacteria leads to nucleosome repositioning, which correlates with altered transcription. Nucleosome remodeling follows distinct patterns of nucleosome repositioning at different loci. Using a reverse genetic screen, we identify multiple chromatin remodeling ATPases with previously undescribed roles in immunity, including *EMBRYO SAC DEVELOPMENT ARREST 16*, *EDA16*. Functional characterization of the immune-inducible chromatin remodeling ATPase *EDA16* revealed a mechanism to negatively regulate immunity activation and limit changes in redox homeostasis. Our transcriptomic data combined with MNase-seq data for *EDA16* functional knock-out and over-expressor mutants show that *EDA16* selectively regulates a defined subset of genes involved in redox signaling through nucleosome repositioning. Thus, collectively, chromatin remodeling ATPases fine-tune immune responses and provide a previously uncharacterized mechanism of immune regulation.

## Author summary

Immune signaling is tightly controlled to avoid inappropriate activation leading to severe developmental penalties. Following the perception of microbes, multiple signaling cascades are initiated leading to transcriptional activation of immunity. The amplitude of

**Funding:** AJP was funded by the University of Warwick through the Biotechnology and Biological Sciences Research Council (BBSRC, https://bbsrc.ukri.org) Midlands Integrative Biosciences Training Partnership (BB/M01116X/1). ADF was supported by the BBSRC/EPSRC funded Warwick Integrative Synthetic Biology Centre (BB/M017982/1, https://bbsrc.ukri.org) awarded to VN. Work in VN laboratory is supported by the Royal Society (UF160546, https://royalsociety.org). SHS was supported by a European Research Council (ERC, https://erc.europa.eu) under the European Union's Horizon 2020 research and innovation programme (grant agreement No 678511). LF was supported by a BBSRC Discovery Fellowship. The funders had no role in study design, data collection and analysis, decision to publish, or preparation of the manuscript.

**Competing interests:** The authors have declared that no competing interests exist.

this response can be regulated at multiple levels, including chromatin. Here we show that activation of plant immunity affects nucleosome positioning over thousands of loci and is correlated with the transcription of immune-related genes. A reverse genetic screen of chromatin remodeling ATPases identified six genes with novel roles in plant immunity. We further characterize the role of *EDA16* as a negative regulator of immune responses. *EDA16* expression is induced upon activation of immunity and regulates a subset of genes involved in redox homeostasis through nucleosome repositioning.

## Introduction

Plant leaf surfaces are inhabited by diverse microbial communities [1]. Remarkably, plants are resilient to most microbial infections and disease is the exception. The success of plant defenses relies on physical barriers and a sophisticated, multi-layered, highly tunable immune system capable of precisely assessing and responding to the various threats encountered in nature [2]. Plasma membrane localized pattern recognition receptors (PRRs) detect microbe-associated molecular patterns (MAMPs) such as the bacterial flagellin (or its active peptide epitope flg22). PRRs initiate a signaling cascade leading to MAMP-triggered immunity (MTI). Early MTI responses include rapid production of reactive oxygen species (ROS), calcium influx, activation of mitogen-activated protein kinases (MAPKs) and differential regulation of gene expression of approximately 10 per cent of the plant genome [3,4]. These collective MTI responses are sufficient to ward off most microbes. However, adapted pathogens can cause disease primarily by employing effector proteins capable of attenuating MTI or altering plant cell signaling in their favor [2]. In an evolutionary arms race, plants have in turn evolved cytoplasmic resistance (R) proteins that detect the presence of pathogen-derived effectors. R proteins initiate effector-triggered immunity (ETI), a strong immune response that often results in localized cell death to limit the growth and spread of the pathogen [2]. Importantly, components of both MTI and ETI have been successfully employed to improve crop disease resistance [5,6].

Activation of plant immunity often comes with severe developmental penalties, most notably reduced growth and yield [7]. Therefore, plant immune responses must be tightly controlled. Given the plethora of microbes associated with plants, it is not surprising that MTI is heavily regulated to enable the optimal amplitude of immune responses and to terminate signaling once the pathogen threat is over. Numerous phosphatases have been shown to associate with PRRs and act as regulators of MTI [8] or to control transduction of downstream signaling [9,10]. Other proteins acting as regulators of MTI include E3 ligases [11,12], and MAPKs [13] among many others.

Plant immune responses are also controlled at the chromatin level where DNA methylation, histone modifications and chromatin remodeling complexes play crucial regulatory roles [14]. Chromatin remodeling complexes evict, slide or reposition nucleosomes around DNA through the action of their core component, the chromatin remodeling ATPase [15]. The chromatin remodeling ATPases *SPLAYED* (*SYD*) and *BRAHMA* (*BRM*) regulate the expression of several defense-related genes [16]. *SYD* also regulates a subset of genes involved in response to the immune-associated jasmonic acid (JA) and ethylene (ET) hormonal pathways [17]. Other chromatin remodeling ATPases, such as *PHOTOPERIOD-INDEPENDENT EARLY FLOWERING 1* (*PIE1*), and *DECREASE IN DNA METHYLATION1* (*DDM1*) are associated with gene silencing and negative regulation of plant defense responses [18–20]. In addition, *DDM1* affects the expression of the *SUPPRESSOR OF NPR1-1 CONSTITUTIVE 1*

(*SNC1*) *R* gene, a constitutive repressor of *PATHOGENESIS-RELATED GENE 1* (*PR1*) [21]. The chromatin remodeling ATPase *CHR5* functions antagonistically to *DDM1* as a positive regulator of *SNC1* expression [22]. Furthermore, the rice chromatin remodeling ATPase *BRHIS1* constitutively represses defenses in a salicylic acid (SA)-independent manner [23]. Thus, remodeling of chromatin, and particularly chromatin remodeling ATPases, play essential roles in orchestrating plant immune gene expression.

Despite the evidence implicating multiple chromatin remodeling ATPases in gene regulation during biotic stress, the impact of MTI on chromatin dynamics and associated gene expression regulation remains largely unexplored. Here, using micrococcal nuclease digestion and mono-nucleosome DNA purification followed by Illumina sequencing (MNase-seq) paired with RNA-seq, we reveal the effects of MTI activation on nucleosome repositioning and its correlation with flg22-regulated transcriptional changes. Moreover, by performing a comprehensive reverse genetic screen, we were able to identify several chromatin remodeling ATPases that modulate plant immunity. We characterized in detail the ATPase *EMBRYO SAC DEVELOPMENT ARREST 16*, *EDA16*, and show that it functions as an MTI-induced regulator of cellular redox homeostasis during immune responses.

## Results

### Activation of MTI leads to nucleosome repositioning at specific loci

Activation of MTI causes substantial transcriptional reprogramming [3,4], but its effect on chromatin remodeling remains unclear. First, we explored the effect of flagellin (flg22) on nucleosome remodeling at the single cell level using GFP-tagged histone H2B fluorescence recovery after photobleaching (FRAP) as a proxy for nucleosome dynamics [24]. Interestingly, we found that both in *Arabidopsis* as well as *Nicotiana benthamiana*, the presence of flg22 led to a faster FRAP recovery, suggesting an increased nucleosome remodeling status associated with flg22 perception (S1A and S1B Fig).

In order to investigate the MTI-induced DNA-nucleosome dynamics and their influence on transcriptional changes, we conducted MNase-seq in parallel with RNA-seq experiments. MNase digests DNA unprotected by nucleosomes, allowing for mono-nucleosome DNA isolation and next-generation sequencing. *Arabidopsis thaliana* Col-0 (wild type) seedlings were treated with 100 nM flg22 or water (mock-treatment) for 2 hours. We identified 2612 Differentially Expressed Genes (DEGs, adjusted p-value < 0.05, fold-change > 1.5) in response to flg22 treatment (S1C Fig and S1 Dataset). Over 80% of these DEGs were also identified in a recent study using similar conditions [25], validating our results. In parallel, the MNase-seq experiment, with ~48 million reads per replicate and a 28-fold coverage on average (S1 Table), identified a nucleosome phase, both in mock and flg22-treated samples, of approximately 177 base pairs (bp) between nucleosome peaks with no statistical differences between mock and treated samples (S1D Fig), paired T-test p-value > 0.01. This result is in line with previous findings for *Arabidopsis* mature leaves: 185 bp, and flowers: 182 bp [26] and supports the notion that neither developmental stage nor activation of immunity, change the average genomic nucleosome distribution.

To statistically assess the dynamic changes attributable to the flg22 response at the nucleosome level, we used DANPOS (Dynamic Analysis of Nucleosome Position and Occupancy by Sequencing), a tool specifically designed to determine dynamic changes of nucleosome position associated with environmental changes. DANPOS analyses changes in three categories of nucleosome dynamics; location, fuzziness, and occupancy [27]. These parameters refer to changes in peak intensity, differences in broadness of peak or shifts from their reference position, respectively. Our analysis identified 659,053 nucleosome peaks, of which 27,102 (~ 4%)

were differentially positioned nucleosomes (DPNs, FDR < 0.01) between mock and flg22 elicitation in at least one of the three parameters compared by DANPOS (S1E and S1F Fig and S2 and S3 Tables and S2 and S7 Datasets). We then mapped these DPNs to protein-coding genes with 1,000 nucleotides upstream from their Transcription Start Sites (TSS), considered as promoter regions. We identified 13,938 (S3 Table, S2 Dataset, S7 Dataset) genes containing one or more DPNs, of which about 10% (1384) overlapped with DEGs. Amongst these 1384 DEGs, 1,142 were flg22-induced, which is more than what would be expected from a random overlap (hypergeometric p-value < 0.01), and 242 were flg22-repressed genes showing no statistical over-representation (hypergeometric p-value > 0.01). Collectively, more than half of the flg22-DEGs identified by RNA-seq contained altered nucleosome patterns (1,142 induced genes and 242 repressed genes). Both flg22-induced and repressed genes with DPNs were enriched for Gene Ontology (GO) terms associated with infection and response to pathogens (Fig 1A and S3 Dataset). Surprisingly, most of the genes with DPNs (~90%) were not DEGs following elicitation with flg22 (12,554). However, non-DEGs with DPNs were enriched for GO terms involved in growth arrest, early flowering or chromatin remodeling (Fisher's Exact Test, p-value < 0.01), suggesting that transcription of these genes may be poised for future alterations (Fig 1A and S4 Dataset). The average nucleosome occupancy profiles remained similar across conditions (Fig 1B). However, analysis of differential nucleosome occupancy revealed that, on average, there is a nucleosome depletion across the gene bodies of flg22-induced genes with DPNs in response to flg22 treatment (Fig 1C). This contrasted with the average trend for flg22-repressed genes with DPNs, which showed an increase of nucleosome occupancy over their gene bodies (Fig 1C and S5 Dataset). Overall, our results are in agreement with previous work in *Arabidopsis* showing that higher nucleosome occupancy correlates with lower gene expression [28].

It is well established that nucleosome positioning fluctuates in several distinct ways which can affect gene expression [29]. To separate different effects, we used K-means clustering to further dissect the flg22 response at the chromatin level. We focused on the 1142 flg22-induced genes with DPNs as a subset of genes sufficiently large for appropriate clustering analysis (Fig 2 and S6 Dataset). Clusters 1, 2 and 3, containing two thirds of the flg22-induced genes with DPNs (762), showed a decrease in nucleosome occupancy along the gene body, at the +2 nucleosome and promoter regions respectively. Interestingly, clusters 4, 5 and 6 showed the opposite trend with an increase in nucleosome occupancy at nucleosome +2, +1 and -1 respectively, suggesting that a few specific well-positioned nucleosomes can be crucial for the induction of gene expression. Genes within cluster 3 had a distinct reduction of nucleosome occupancy approximately 500 bp upstream from the +1 nucleosome following elicitation with flg22, hinting that these genes might be under the regulation of TFs requiring nucleosome-free regions conditional to plant immune responses. Taken together, our results show that flg22 elicitation alters nucleosome positioning in the promoters and gene bodies of a large number of genes and demonstrate that distinct flg22-induced nucleosome repositioning correlates with transcription induction.

## Chromatin remodeling ATPase mutants present altered immune responses

The observed flg22-dependent nucleosome repositioning can be accounted for by several factors, including the action of chromatin remodeling ATPases [15]. *Arabidopsis*, as all land plants, possesses a large family of chromatin remodeling ATPases, as identified by the conserved N-terminal SNF and C-terminal HELIC-domains [30]. In order to investigate the potential role of these chromatin remodeling ATPases in plant immunity, we chose 20 genes covering half of the family for functional characterization using mutant analysis, paying special

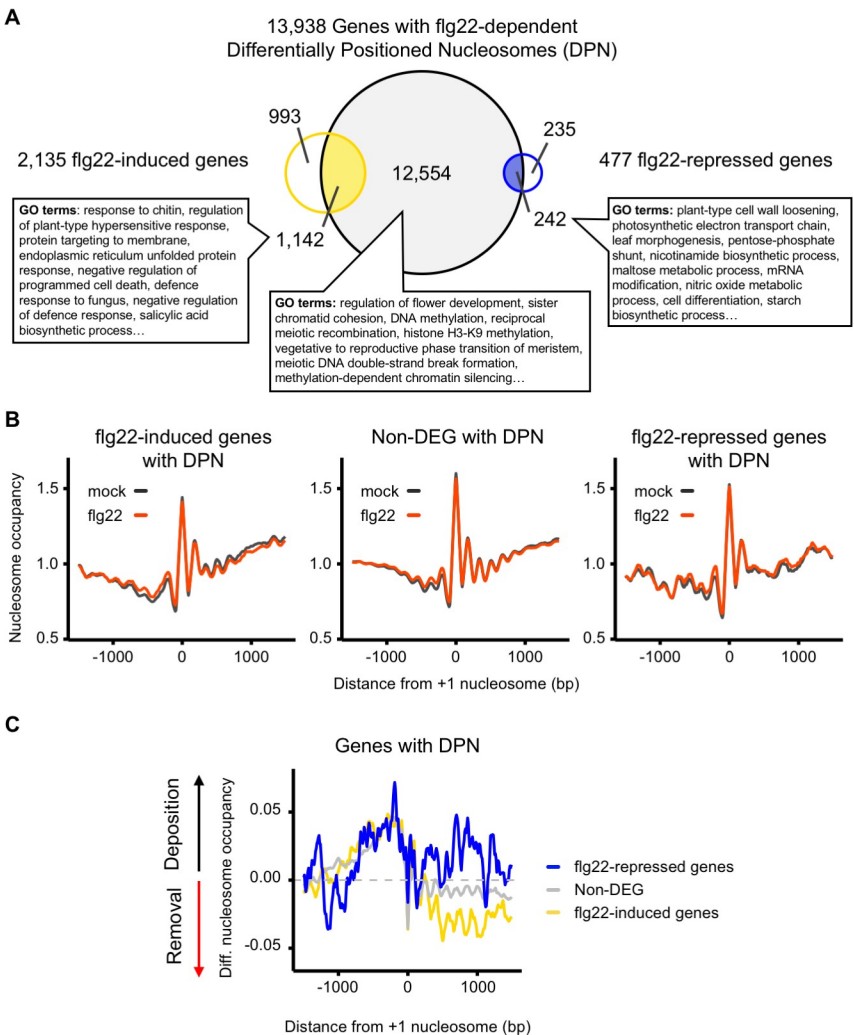

**Fig 1. Activation of MTI results in nucleosome repositioning that correlates with gene expression.** (A) flg22 elicitation results in Differentially Positioned Nucleosomes (DPN). 2-week-old Col-0 seedlings were treated for 2 hours with 100 nM flg22 before harvesting for RNA-seq and MNase-seq analysis. Venn diagram illustrating the overlap between genes (protein-coding genes plus 1000 nucleotides upstream their Transcription Start Sites, TSS) with at least one DPN (grey), flg22-induced genes (yellow), and flg22-respressed genes (blue). Most significant GO terms found for the intersection groups with the TopGO package using as a control set all Arabidopsis protein coding genes (Fisher Exact Test, p-value < 0.01). (B) Changes in nucleosome occupancy in the promoters and the gene bodies following flg22 elicitation. Average nucleosome occupancy detected with MNase-seq analysis, mock (black) and flg22 (red) for flg22-induced genes with DPNs (left panel), Non-Differentially Expressed Genes (Non-DEGs) with DPNs (middle panel) and flg22-repressed genes (right panel). Graphs are centred on the +1 nucleosome from the gene TSS. (C) Differential nucleosome occupancy following flg22 elicitation. Average of the nucleosome occupancy differences between flg22- and mock treatment of flg22-induced (yellow), Non-DEGs (grey), and flg22-repressed genes (blue) for genes with DPN. The graph is centred on the +1 nucleosome from the gene TSS.

attention to uncharacterized genes or those not yet associated with plant immunity (Tables 1 and S4). Five-week-old homozygous T-DNA insertion mutant plants were spray-inoculated with a *Pseudomonas syringae* pathovar *tomato* DC3000 (*Pst* DC3000) strain lacking the effectors *AvrPto* and *AvrPtoB* (*Pst* DC3000 *ΔavrPtoΔavrPtoB*) in order to discern milder phenotypes. Of the 20 chromatin remodeling ATPase mutants tested, six showed altered immune responses (Table 1). In comparison with Col-0 control plants, *PICKLE RELATED 2* (*PKR2*, also known as *CHR7*) and *RAD54* (also known as *CHR25*) mutants were more susceptible to

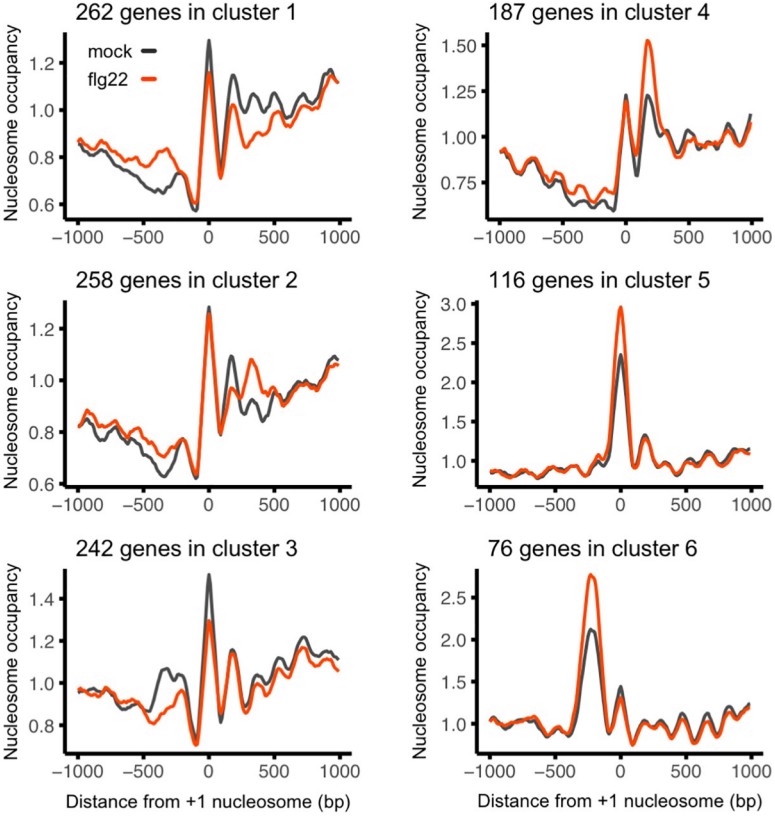

**Fig 2. flg22-induced changes in nucleosome remodeling follow distinct patterns of nucleosome repositioning.** K-means clustering of differential nucleosome occupancy. 1,142 flg22-induced genes with Differentially Positioned Nucleosomes (DPNs) were clustered in 6 groups with marked differences in average nucleosome occupancy between flg22 elicitation (red) and mock treatment (black). The graph is centred on the +1 nucleosome from the gene TSS.

*Pst* DC3000 *ΔavrPtoΔavrPtoB*. By contrast, *CHROMATIN REMODELING FACTOR 17* (*CHR17*), *CHROMATIN REMODELING 19* (*CHR19*, also known as *ETL1*), and two genes within the Ris1 subfamily, *SNF2- RING-HELICASE–LIKE 2* (*FRG2*, also known as *CHR28*), and *EMBRYO SAC DEVELOPMENT ARREST 16* (*EDA16* also known as *FRG4*) presented enhanced disease resistance phenotypes (Table 1).

### *EDA16* attenuates MTI responses

From the six genes of chromatin remodeling ATPases that mutation lead to altered disease phenotypes, *EDA16* had the highest flg22-induced expression according to our RNA-seq data (Table 1). In agreement, publicly available transcriptomic databases indicated that flg22, *Pst* DC3000 infection, oxidative stress such as hydrogen peroxide ($H_2O_2$) and ozone exposure result in the induction of *EDA16* expression [31]. Our independent validation by qPCR further confirmed that *EDA16* gene expression was induced within 1 hour after flg22 elicitation (Fig 3A) and within 3 hours and onwards following infection with *Pst* DC3000 (Fig 3B). We therefore focused on this chromatin remodeling ATPase since its role in immunity has not been previously characterized.

To better understand the immune phenotypes caused by mutation of the *EDA16*, we tested two additional homozygous T-DNA insertion mutants. The original mutant used in our screen (*SAIL_735_G06*) has a T-DNA insertion towards the 3' end of the *EDA16* gene, within the

**Table 1. Bacterial susceptibility screening of chromatin remodeling ATPase mutants.**

| | | Gene | | FC | Bacterial growth |
|---|---|---|---|---|---|
| Rad5 | AT1G08060 | | *MOM1* | 0.97 | **No dif.** |
| | AT3G16600 | | *FRG3* | 1.12 | **No dif.** |
| | AT3G54460 | | | 0.92 | **No dif.** |
| | AT1G05120 | | | 1.15 | **No dif.** |
| | AT1G02670 | | | 1.13 | - |
| | AT2G40770 | | | 0.85 | - |
| | AT5G05130 | | | 0.85 | - |
| | AT5G22750 | | *RAD5* | 0.89 | - |
| | AT5G43530 | | | 0.94 | - |
| Snf2 | AT4G31900 | | *CHR7, PKR2* | 0.98 | **Susceptible** |
| | AT2G25170 | | *CHR6, CHD3* | 0.85 | - |
| | AT5G44800 | | *CHR4, PKR1* | 0.99 | - |
| | AT2G13370 | | *CHR5* | 1.04 | No dif. (22) |
| | AT2G46020 | | *CHR2, BRM* | 1.06 | - |
| | AT2G28290 | | *CHR3, SYD* | 0.94 | No dif. (17) |
| ISWI | AT5G18620 | | *CHR17* | 0.99 | **Resistant** |
| | AT3G06400 | | *CHR11* | 1.06 | - |
| Snf1 | AT5G19310 | | *CHR23, MINU2* | 0.95 | - |
| | AT3G06010 | | *CHR12, MINU1* | 0.91 | - |
| | AT5G66750 | | *CHR1, DDM1* | 0.93 | Resistant (20, 21) |
| | AT2G44980 | | *CHR10, ASG3* | 1.29 | **No dif.** |
| | AT2G02090 | | *CHR19, ETL1* | 1.01 | **Resistant** |
| | AT3G12810 | | *CHR13, PIE1* | 1.03 | Resistant (18, 19) |
| Ino80 | AT3G57300 | | *INO80* | 0.87 | - |
| | AT3G54280 | | | 0.99 | - |
| | AT1G48310 | | *CHR18* | 0.96 | **No dif.** |
| | AT5G07810 | | | 0.87 | **No dif.** |
| Rad54 | AT1G03750 | | *CHR9, SWI2* | 1.08 | **No dif.** |
| | AT3G19210 | | *CHR25, RAD54* | 1.48 | **Susceptible** |
| | AT1G08600 | | *CHR20, ATRX* | 0.88 | **No dif.** |
| | AT2G18760 | | *CHR8* | 1.20 | **No dif.** |
| | AT5G63950 | | *CHR24* | 0.96 | - |
| | AT2G21450 | | *CHR34* | 0.89 | **No dif.** |
| | AT5G20420 | | *CHR42* | 0.86 | - |
| | AT3G42670 | | *CHR38, CLSY1* | 0.93 | - |
| | AT3G24340 | | *CHR40* | 1.03 | - |
| | AT1G05490 | | *CHR31* | 0.84 | **No dif.** |
| Ris-1 | AT3G20010 | | *CHR27, FRG1* | 0.86 | **No dif.** |
| | AT1G50410 | | *CHR28, FRG2* | 1.17 | **Resistant** |
| | AT1G61140 | | *EDA16, FRG4* | **1.57** | **Resistant** |
| | AT1G11100 | | *FRG5* | **3.66** | **No dif.** |

*A. thaliana* chromatin remodeling ATPases are sorted by protein phylogeny. Gene names and alternative names, if known, are indicated. The Fold-Change (FC) in gene expression from RNA-seq analysis for indicated genes upon elicitation of 2-week-old Col-0 plants with 100 nM flg22 (red; adjusted p-value < 0.05 and FC > 1.5). For each chromatin remodeling ATPase gene, two T-DNA insertion mutant plants and Col-0 (control) at the 5-week-old stage were spray-inoculated with *Pst* DC3000 *ΔavrPtoΔavrPtoB*. Bacterial colony-forming units in Col-0 control plants and the indicated mutants were determined 3 days post-infection. Based on bacterial growth, mutants were characterized as susceptible, resistant or having no differences in comparison with Col-0 plants (red for statistically significant differences with a two-sided T-test, p-value < 0.05, n = 6. – for genes not tested). Previously described phenotypes affecting immunity are indicated.

region encoding for the conserved HELICc domain. One of the additional mutants has a T-DNA insertion within the promoter region (*SAIL_40_F09*) while another contained an insertion within the coding region for the SNF domain (*SALK_208691*) (Fig 3C). Next, we examined *EDA16* cDNA integrity and gene expression levels in all three mutants. Mutant *SAIL_735_G06* T-DNA insertion disrupts the conserved HELICc domain, essential for the catalytic activity and function of chromatin remodeling ATPases in plants [32] and other organisms [33]; we therefore referred to it as *eda16–ΔHELICc* (*eda16-ΔHc*). Similarly, the *SALK_208691* mutant also produced a truncated *EDA16* mRNA (Fig 3C and 3D). In contrast, the *SAIL_40_F09* (promoter-located) mutant showed no transcript disruption (Fig 3D) but it had higher transcript level than Col-0 (Fig 3E). Following elicitation with flg22 the *SAIL_40_F09* mutant showed significantly higher expression levels of *EDA16* in comparison with Col-0 plants (Fig 3F). We therefore refer to the *SAIL_40_F09* mutant as *eda16–OVER-EXPRESSOR* (*eda16-OE*) hereafter. Despite the differences in *EDA16* expression prior to elicitation (Fig 3E), none of the *eda16* mutants showed any obvious growth phenotype during the vegetative stage (Fig 3G), indicating that EDA16 does not have a general role in plant development.

We next tested the immune phenotypes of the three *EDA16* homozygous mutants upon challenge by *Pst* DC3000 and *Pst* DC3000 *ΔavrPtoΔavrPtoB*. The *eda16-OE* mutant showed enhanced susceptibility to both strains of *P. syringae*. In contrast, the truncated mutants *SALK_208691* and *eda16-ΔHc* had enhanced resistance phenotypes (Fig 3H, 3I and 3J), suggesting that *EDA16* is a negative regulator of plant immunity. To gain an understanding of the transcriptional control of *EDA16* (and in particular during immunity onset), we performed an *in silico* transcription factor binding site motif analysis over 1000 bp of *EDA16* promoter using TRANSFAC publicly available dataset [34]. Among other common transcription factor binding sites such as MIB and ABI sites, we found a WRKY18 binding site. The availability of ChIP-seq data for this TF allowed us to corroborate the presence of a peak within the expected region (~300–600 bp upstream *EDA16* TSS). Consistent with this analyis, *wrky18* mutant transcriptional data showed no upregulation of *EDA16* 2 hours after flg22 exposure where the same dataset showed a clear induction for *EDA16* on Col-0 [25].

In order to clarify the role of *EDA16* in early MTI signaling, we monitored the expression of well-characterized MTI-induced marker genes regulated by the MAPK (i.e. *FRK1*), the CDPK (i.e. *PHI-1*) and SA signaling pathways (i.e. *CBP60g*). The induction of all marker genes was indistinguishable between the *eda16* mutants and Col-0 control plants (S2A Fig), demonstrating that early MTI transcriptional activation is not affected by the *EDA16* mutation.

Given the widespread recognition of *Pst* DC3000 effectors by *Arabidopsis* [35], we next tested the role of *EDA16* in ETI. Adult plants were syringe-infiltrated with *Pst* DC3000 expressing the avirulent effector *AvrRpt2*, which is recognized by the *Arabidopsis* resistance protein RPS2 [36]. Ion leakage assays (indicative of cell death) showed no difference between *eda16* mutants and Col-0 control plants (S2B Fig), suggesting that ETI is not compromised in the mutant plants. Bacterial growth assays further supported this conclusion, where *eda16* mutants and Col-0 control plants displayed indistinguishable immune phenotypes against *Pst* DC3000 *avrRpt2* (S2C Fig). Thus, *EDA16* is not involved in early MTI or ETI responses. Taken together, these results support a model where *EDA16* is upregulated following activation of MTI by *Pst* DC3000, in order to attenuate MTI and enable the optimal amplitude of responses.

## *EDA16* alters nucleosome positioning and expression of flg22-regulated genes

Our next objective was to identify the role of *EDA16* in flg22-induced nucleosome repositioning. Following activation of MTI, *EDA16* expression peaked at approximately 2–3 hours post

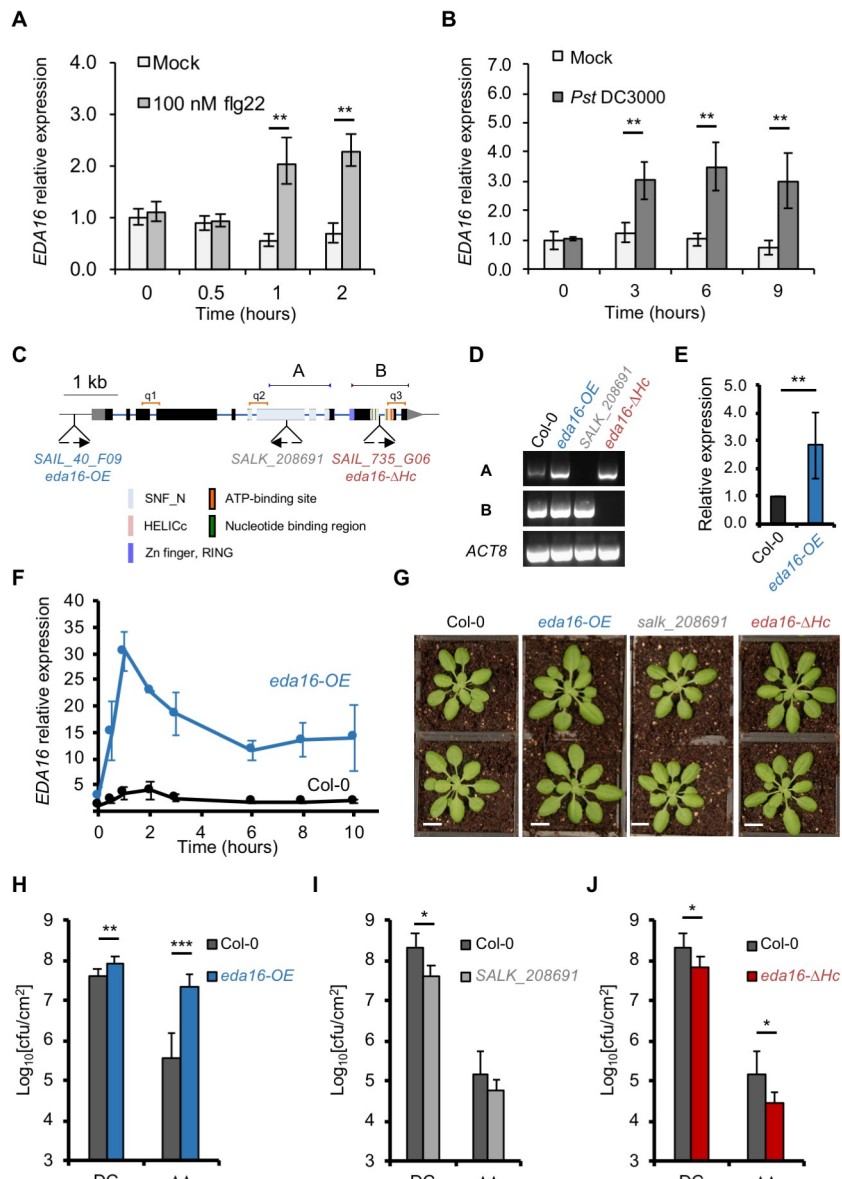

**Fig 3. *EDA16* is a negative regulator of plant immunity.** (A) *EDA16* expression is induced by flg22 elicitation. Accumulation of *EDA16* transcript was assessed by qPCR in 2-week-old Col-0 seedlings elicited with 100 nM flg22 or water (mock). Values are average of three biological repeats ± SE presented as fold induction compared with mock-treated sample at time 0. (B) Bacterial infection induces *EDA16* expression. 5-week-old Col-0 plants were infiltrated with *Pst* DC3000 or 10 mM MgCl$_2$ (mock). *EDA16* expression was assessed by qPCR. Values are average of three biological repeats ± SE presented as fold induction compared with mock-treated sample at time 0. Labelled values are statistically different as established by two-sided T-test p-values: ** < 0.01. (C) Schematic representation of the T-DNA insertions in EDA16 gene. Boxes and solid lines denote exons and introns, respectively. T-DNA insertions and mutant names are indicated below the gene structure. The different functional domains of EDA16 are color-coded. Primers used for RT-PCR presented in panel D and corresponding PCR products are indicated above the gene structure (**a, b**). qPCR primers used in panels E and F are indicated above the gene structure (q1, q2, q3). (D) Mutant characterization by cDNA integrity. RT-PCR analysis of *EDA16* gene expression in homozygous *eda16* mutants and Col-0 plants. The amplified fragments (**a** and **b**) are indicated in C. *ACT8* was used as a control. (E) *SAIL_40_F09* mutant is an of *EDA16* over-expresser. Accumulation of *EDA16* transcript was assessed by qPCR in 2-week-old Col-0 and *SAIL_40_F09* (*eda16-OE*) by averaging the results of 3 primer pairs (q1, q2 and q3), presented in panel C. Values are average of three biological repeats ± standard deviation presented as fold induction compared with Col-0 at time 0. (F) The *SAIL_40_F09* mutant is an inducible over-expresser of *EDA16*. Accumulation of *EDA16* transcript was assessed by qPCR in 2-week-old Col-0 and *SAIL_40_F09* (*eda16-OE*) mutant plants as in panel E, after elicitation with 100 nM flg22 at the indicated times. Values are average of three biological repeats ± standard deviation presented as fold

induction compared with Col-0 at time 0. (G) Representative pictures of 5-week-old *eda16* mutants and Col-0 plants (bar = 1 cm). (H, I and J) The *eda16* knock-out and over-expresser mutants have opposite immunity phenotypes. 5-week-old Col-0 (black), *eda16-OE* (blue), *salk_208691* (grey) and *eda16-ΔHc* (red) plants were spray-inoculated with *Pst* DC3000 (DC) and *Pst* DC3000 *ΔavrPtoΔavrPtoB* (ΔΔ) as indicated. Bacterial numbers were determined 3 days post-infection. Error bars represent standard deviation (n = 6). The experiment was repeated 3 times with identical results. Labelled values are statistically different as established by two-sided T-test p-values: * < 0.05, ** < 0.01, *** < 0.001. Cfu stands for colony-forming units.

elicitation (Fig 3A, 3B and 3F). We, therefore, conducted MNase-seq and RNA-seq experiments on Col-0, *eda16-OE* and *eda16-ΔHc* plants 2 hours after elicitation with flg22 or mock treatment. Nucleosome phasing was not altered in the *eda16* mutants following elicitation with flg22 (S3A Fig), suggesting that, unlike the ISWI subfamily of chromatin remodeling ATPases [37], EDA16 is not involved in orchestrating genome-wide nucleosome spacing and general maintenance of chromosome structure. In contrast, analysis of nucleosome dynamics by DANPOS detected an increased number of flg22-dependent DPNs in the *eda16-OE* mutant (28,796) and a decrease in the *eda16-ΔHc* mutant (21,386) compared to Col-0 (27,102), (S2 Table and S7 Dataset). Mapping these DPNs to protein-coding genes plus 1000 nucleotides upstream of their TSS, revealed that flg22-dependent DPNs occurred at both overlapping and distinct loci for Col-0 and the *eda16* mutants (Fig 4A and S2, S9 and S11 Datasets). Most importantly, differential nucleosome occupancy analysis of flg22-induced genes with DPNs revealed opposite trends for *eda16-OE* and *eda16-ΔHc* mutants (Fig 4B). In comparison with Col-0 (control), the *eda16-OE* mutant displayed increased nucleosome occupancy at the promoter regions, whereas decreased occupancy was observed over the gene bodies. In contrast, the *eda16-ΔHc* mutant had a noticeable decrease in nucleosome occupancy at the promoter regions (Fig 4B). Our results show that upon activation of MTI, flg22-induced genes have distinct nucleosomes densities in excess (*eda16-OE*) or functional absence (*eda16-ΔHc*) of *EDA16*, supporting the notion that EDA16 regulates MTI through changes in nucleosome occupancy.

In parallel, the RNA-seq data enabled us to ask if there was a correlation between the observed differences in nucleosome occupancy and changes in gene expression. Principal component analysis (PCA) of the gene expression levels showed that the majority of the variance, nearly 80%, could be attributed to activation of MTI by flg22 elicitation (S3B Fig). In agreement, Col-0 and the two *EDA16* mutants shared approximately 60% of the DEGs following activation of MTI (Fig 4C and S8, S10 and S13 Datasets). Despite the significant overlap of flg22-DEGs, there were quantitative differences between Col-0 plants and the two *eda16* mutants. In comparison with Col-0 control plants, the *eda16-OE* mutant showed reduced flg22-dependent induction and the *eda16-ΔHc* mutant an enhanced flg22-dependent induction of gene expression based on the behavior of 2,135 flg22-induced genes (S3C Fig), corroborating that *EDA16* has a role in negatively regulating flg22-mediated gene expression. Furthermore, RNA-seq sample separation by genotype (PC3 and PC4), accounted for ~13% of the variance (S3B Fig). Taken together, these results fit with a model where *EDA16* regulates nucleosomes deposition at the promoters and gene bodies of flg22-induced genes to moderate the expression of a subset of these genes.

## The *EDA16* mutation alters oxidative stress-related gene expression and cellular redox state

To discern genes directly regulated by EDA16 upon activation of MTI, we searched for genes with different expression levels and distinct nucleosome densities in Col-0, *eda16-OE* and *eda16-ΔHc* seedlings following elicitation with flg22. We compared pairwise the expression of

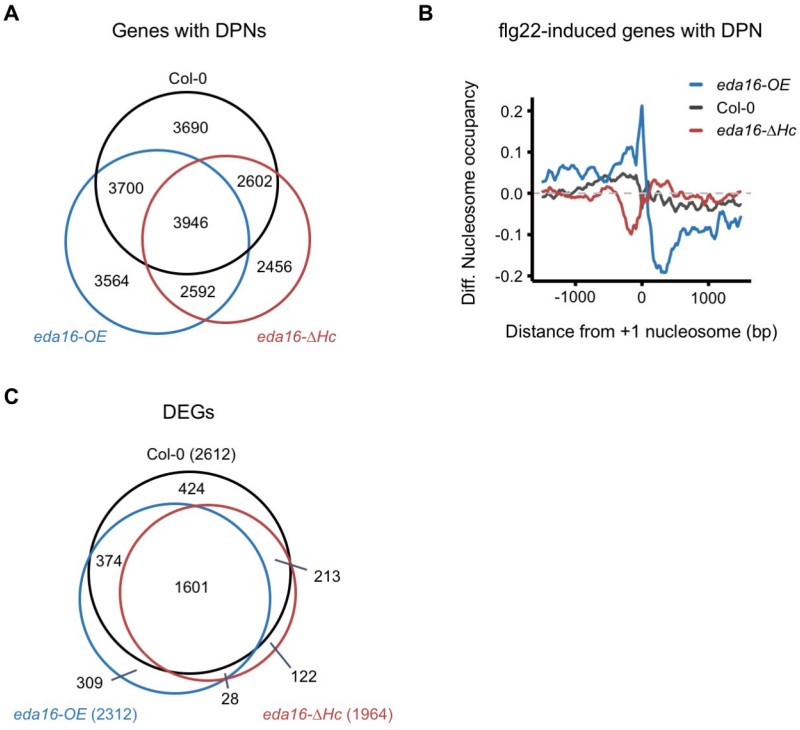

**Fig 4. EDA16 alters nucleosome positioning and expression of flg22-regulated genes.** (A) The *EDA16* mutation alters flg22-induced nucleosome positioning. Venn diagram illustrating the overlap between genes (protein-coding genes plus 1000 nucleotides upstream their Transcription Start Sites, TSS) with at least one Differentially Positioned Nucleosome (DPN) in Col-0 (black), *eda16-OE* (blue) and *eda16-ΔHc* (red) 2 hours after elicitation with 100 nM flg22. (B) flg22-induced genes have distinct nucleosome occupancies in the *eda16-OE* and *eda16-ΔHc* mutants. Average of the nucleosome occupancy differences between flg22-treated and mock-treated Col-0 (black), *eda16-OE* (blue), and *eda16-ΔHc* (red) for flg22-induced genes. The graph is centred on the +1 nucleosome from the gene TSS. (C) The effect of *EDA16* mutation on the flg22 response at the transcriptomic level. Venn diagram illustrating the overlap between flg22-regulated, Differentially Expressed Genes (DEGs) in Col-0 (black), *eda16-OE* (blue) and *eda16-ΔHc* (red) 2h after elicitation with 100 nM flg22.

flg22-DEGs between Col-0 and *eda16-OE*, Col-0 and *eda16-ΔHc* and between *eda16-OE* and *eda16-ΔHc*. Our analysis identified 21 genes with quantitatively different expression levels between Col-0 and the *eda16* mutants that are also differentially regulated between the two mutants (Fig 5A). Consistent, with our previous analysis (Fig 4B), the identified genes had distinct nucleosome densities in their promoters and gene bodies in the *eda16-OE* and *eda16-ΔHc* mutants. In comparison with Col-0 (control), the *eda16-OE* mutant displayed increased nucleosome occupancy at promoter regions and decreased occupancy over gene bodies. In contrast, in the *eda16-ΔHc* mutant, there was a noticeable reduction of nucleosome occupancy at the promoters of the 21 genes identified (Fig 5B), suggesting that EDA16 mediates nucleosome repositioning between gene bodies and promoters. Out of these 21 genes, 10 showed a clear pattern of a compromised flg22 induction in the *eda16-OE* mutant and an exaggerated flg22 induction in the *eda16-ΔHc* mutant (Fig 5A). GO term analysis for this group of genes showed enrichment in response to high light, hydrogen peroxide and heat acclimation (Fig 5A and S12 Dataset), all of which involve extensive changes in cellular redox homeostasis.

In order to explore the molecular signaling underlying the cellular homeostasis changes in the mutants, we examined total glutathione (GSH) levels since GSH is known to control cellular redox. Glutathione is a small molecule with multiple functions in plants including regulation of immune responses, defense-detoxification and general redox homeostasis [38]. Prior to

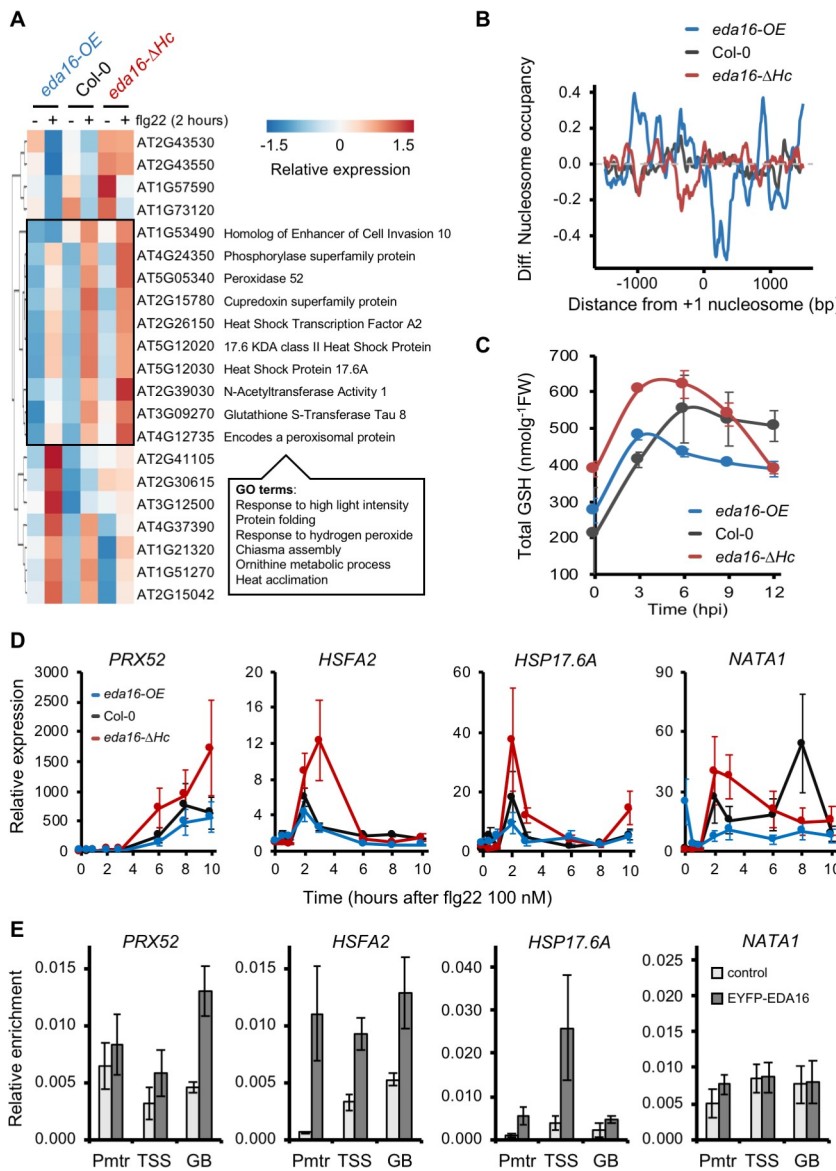

**Fig 5. EDA16 regulates plant redox homeostasis during immune responses.** (A) *EDA16* affects the expression of a subset of flg22-regulated genes. Heatmap of Differentially Expressed Genes between Col-0, *eda16-OE* and *eda16-ΔHc* plants 2h after elicitation with 100 nM flg22. The box on the heatmap indicates genes with a distinct pattern of misregulation in the *eda16-OE* and *eda16-ΔHc* mutant plants and accompanied with their (TAIR10) gene description. Most significant GO terms found for the intersection group. (B) Differential nucleosome occupancy of the 21 *EDA16*-flg22 DEGs. Differences between the average nucleosome occupancy of *EDA16*-regulated flg22-induced genes in Col-0 (grey), *eda16-OE* (blue), and *eda16-ΔHc* (red) 2 hours after elicitation with 100 nM flg22 and mock. The graph is centered on the +1 nucleosome from the gene TSS. (C) The EDA16 mutation alters glutathione concentration. Total glutathione (GSH) levels as concentration per fresh weight were measured in 3-week-old Col-0 (black), *eda16-OE* (blue) and *eda16-ΔHc* (red) plants at the indicated times following infection with *Pst* DC3000. Error bars represent standard deviation, n = 3. (D) *EDA16* negatively regulate the expression of target genes. Gene expression of *PRX52*, *HSFA*, *HSP17.6A* and *NATA1* assessed by qPCR in 2-week-old Col-0 (black) *eda16-OE* (blue) and *eda16-ΔHc* (red) seedlings elicited with 100 nM flg22. Values are average of three biological repeats ± SE presented as fold induction compared with Col-0 mock-treated sample at time 0. (E) EDA16 directly binds on target genes. ChIP-qPCR was performed on leaves from Col-0 and Col-0 *35S::EDA16-YFP* 5-week-old plants (n = 20) to assess EDA16 binding to *PRX52*, *HSFA*, *HSP17.6A* and *NATA1*. Three primer pairs were used for each gene corresponding to promoter region (Pmtr), TSS and gene body (GB). Values are average of three biological repeats ± SEM presented as relative enrichment compare to input.

activation of immunity, the *eda16-ΔHc* mutant exhibited elevated GSH levels in comparison with Col-0 control plants (S4A Fig). More importantly, in comparison with Col-0 control plants during the first 6–9 hours post-infection with *Pst* DC3000, the accumulation of total GSH was enhanced in the *eda16-ΔHc* mutant, while was decreased in the *eda16-OE* mutant plants (Fig 5C). These differences in GSH levels can be partially explained by the differential gene expression of genes involved in glutathione biosynthesis (S4B Fig). Glutathione levels may regulate immune responses [39–41] potentially accounting for the immunity phenotypes of the *eda16-OE* and *eda16-ΔHc* mutants (Fig 3H, 3I and 3J), which could be explained based on their differential accumulation of glutathione during the early stages of infection (Fig 5C). Four genes within this cluster (*PRX52*, *HSFA2*, *HSP17.6A* and *NATA1*) were selected to determine the effect of EDA16 on their expression pattern. A ten hours long time course of flg22-mediated activation was performed in Col-0 and the mutants *eda16-ΔHc* and *eda16-OE* (Fig 5D). As expected, the selected genes were induced in response to flg22 and we could confirm an increased induction in the absence of EDA16 and a reduced response to flg22 in the *eda16-OE* mutant plants for all four genes. Furthermore–with the exception of *PRX52*, where the expression levels were still increasing at the end of the time course for all three genotypes–the absence of EDA16 caused the transient induction of these genes to be prolonged, supporting the notion of a negative feedback mechanism involving EDA16. Next, we tested whether these genes are direct targets of EDA16 by performing ChIP-qPCRs using Col-0 plants expressing YFP-tagged EDA16 under the control of the *35S* promoter (*35S::EDA16-YFP)*. For detecting EDA16 binding, three primer pairs were used along the gene body for each of the four genes tested (S5B Fig). EDA16 binding was verified using at least one of the primer pairs in *PRX52*, *HSFA2* and *HSP17.6A* (Fig 5E), indicating that at least these genes are direct targets of EDA16. Furthermore, these results are consistent with the observed EDA16-mediated nucleosome repositioning in these genes upon MTI activation (S5B Fig).

Our findings illustrate that following activation of MTI, EDA16-mediated nucleosome repositioning negatively regulates the expression of genes involved in redox homeostasis, presenting a plausible mechanism to prevent the protracted activation of immunity.

## Discussion

To protect themselves against pathogens plants have evolved a heavily regulated immune system enabling the optimal amplitude of immune responses. Given that reprogramming of gene expression is a major part of plant immunity [3,4], we set out to understand the role of chromatin dynamics in regulating the expression of immune-related genes. Transcriptional reprogramming of comparable magnitude are induced by hormonal treatments such as SA and JA [42,43]. In the case of SA treatment, the changes in gene expression correlate with nucleosome repositioning, particularly over the promoter region of genes controlled by NON-EXPRESSER OF PR GENES 1 (NPR1) [44]. Conversely, coronatine treatment, a bacterial analogue of JA, does not significantly alter the nucleosome distribution over genes that are transcriptionally responsive to coronatine [28]. In this work, we found that activation of plant immunity by flg22 resulted in a large number of DPNs. The majority of the identified DPNs were in the promoters and gene bodies of non-DEGs suggesting that these genes may be primed for subsequent transcriptional alterations (Fig 1A). However, as in the case of SA treatment, flg22-induced gene expression also correlates with nucleosome repositioning; over half of the flg22-regulated genes displayed altered nucleosome patterns. Differential nucleosome occupancy analysis showed that flg22 elicitation alters nucleosome occupancy in the promoters and gene bodies of flg22-induced and repressed genes in distinct ways (Figs 1C and 2). In line with this, in mammalian cell lines elicitation with the bacterial-derived MAMP lipopolysaccharide

or viral infections resulted in selective nucleosome repositioning correlated with transcription [45,46]. Furthermore, heat shock treatment in budding yeast resulted in nucleosome repositioning during gene activation [29,47]. Although further studies are needed, the growing list of transcriptional perturbation resulting in nucleosome repositioning suggests that this is a widespread mechanism.

Nucleosome repositioning is mediated by multiple factors, including chromatin remodeling ATPases. In mammalian cells, the chromatin remodeling ATPase SWI/SNF mediates viral infection-induced nucleosome repositioning [46] and the chromatin remodeling SWI/SNF (BAF) complex regulates antiviral activities [48]. In plants, multiple chromatin remodeling ATPases are involved in regulating plant immunity including *SYD*, *BRM*, *DDM1*, *CHR5*, and *PIE1* in *Arabidopsis* and *BRHIS1* in rice [14]. Using reverse genetic screening, we identified six additional chromatin remodeling ATPase mutants with altered immune phenotypes. The mutants of *PKR2* and *RAD54* had enhanced susceptibility phenotypes, while *CHR17*, *ETL1*, *FRG2*, and *EDA16* mutants had enhanced resistance phenotypes (Table 1). *PKR2* regulates multiple processes including cold and salt stress tolerance, flowering and hormonal signaling [49,50]. *RAD54* is involved in DNA repair [51], and its enhanced susceptibility phenotype is reminiscent of those of other DNA repair machinery mutants [52]. *CHR17*, together with the closely related gene *CHR11*, are involved in controlling plant development [37,53–55]. *FRG2* and its close homolog *FRG1*, are implicated in RNA-directed DNA methylation [56]. *ETL1* is mechanistically related to *FRG2* and *FRG1* in transcriptional gene silencing through their association with putative histone methyltransferases SUVR1/2 [57]. Noticeably, the rice ortholog of *FRG2*, *BRHIS1* suppresses rice immunity against the rice blast fungus *Magnaporthe oryzae* [23]. Similarly to the rice *BRHIS1* RNAi lines, the Arabidopsis *frg2-1* mutant exhibited enhanced resistance to *Pst* DC3000 *ΔavrPtoΔavrPtoB* (Table 1), suggesting that the function of chromatin remodeling ATPases in immunity is conserved across plant lineages. Nevertheless, the large number of chromatin remodeling ATPase mutants with altered immune phenotypes highlights the complexity underlying the regulation of plant immunity at the chromatin level.

The chromatin remodeling ATPase *EDA16* displayed a somewhat paradoxical behavior, being induced at the transcriptional level upon MTI perception (Fig 3A and 3B), while analysis of the over-expressor and functional knock-out mutants suggested that *EDA16* is a negative regulator of plant immunity (Fig 3H, 3I and 3J). Interestingly, no differences were observed when studying ETI (S2B and S2C Fig), suggesting perhaps that such a mechanism to dampen immune responses can be only observed at the MTI stage when cell death mechanisms characteristic of the ETI response have not been triggered. Therefore, we focused on the *EDA16* since its role in immunity has not been previously characterized. Our analysis showed that *eda16-OE* and *eda16-ΔHc* mutants display opposite trends in nucleosome occupancy at the promoters and gene bodies of flg22-induced genes (Fig 4B). Yet, we only identified a subset of flg22-regulated genes that were differentially expressed between Col-0, *eda16-OE* and *eda16-ΔHc* (Fig 5A). Furthermore, the flg22-induced differential expression of these genes in Col-0 and the two *eda16* mutants correlates with the differences in nucleosome occupancy (Fig 5B). Therefore, the comparison of gene expression levels and nucleosome densities between the excess (*eda16-OE*) or the functional absence (*eda16-ΔHc*) of EDA16 allowed us to identify the immunity-related genes regulated by EDA16. For 10 out of the 21 flg22-dependent, *EDA16*-regulated genes, the expression was negatively regulated in *eda16-OE* and positively regulated in *eda16-ΔHc* mutants. Importantly, the expression pattern of selected genes was indicative of an EDA16-depedent gene repression; the functional absence of EDA16 (*eda16-ΔHc*) led to prolonged induction, supporting the notion of a negative feedback mechanism involving EDA16 (Fig 5D). Furthermore, we confirmed a direct interaction between EDA16 and three of the four tested loci (Fig 5E). The GO term analysis of

these genes showed that following flg22 elicitation *eda16-ΔHc* has exaggerated, and *eda16-OE* understated hydrogen peroxide transcriptional responses (Fig 5A), suggesting that EDA16 feedback regulates chromatin remodeling to modulate specifically redox-mediated immune responses. Cellular redox status is predominantly underpinned by changes in the levels of the redox-active, immune mediator glutathione [58]. Glutathione-deficient *Arabidopsis* mutants were shown to have enhanced susceptibility to *Pst* DC3000 [39] and be impaired in immune responses [59,60]. More recently, the *Ralstonia solana-cearum* effector RipAY was shown to cause glutathione degradation in order to suppress immunity [40,41]. While it is difficult to discern cause and effect relationships, the differences in glutathione levels following infections with *Pst* DC3000 in the *eda16-ΔHc* and *eda16-OE* mutants in comparison with the Col-0 control plants (Fig 5C) help to explain the opposite immunity phenotypes of the mutants (Fig 3H, 3I and 3J).

In summary, our work shows that activation of MTI results in distinct nucleosome repositioning that correlates with changes in gene expression. Moreover, our work reveals a regulatory mechanism by which the chromatin remodeling ATPase EDA16 acts as a negative regulator of flg22-dependent transcriptional responses. Through nucleosome repositioning, EDA16 regulates the expression of a subset of genes involved in redox homeostasis. Functional absence or excess of EDA16 result in misregulation of oxidative stress responses which in turn has a knock-on effect on the expression of glutathione biosynthesis genes and the subsequent accumulation of glutathione. Therefore, our work elucidates how chromatin remodeling fine-tunes immune responses at both transcriptional and molecular levels in order to enable the optimal amplitude of immune responses.

## Materials and methods

### Plant material

*Arabidopsis* and *Nicotiana benthamiana* seeds were sowed on Arabidopsis Mix or F2 compost soil, respectively, with Intercept and stratified for 2 days at 4°C in darkness. Seeds were germinated and grown in an Aralab growth chamber set at a short photoperiod of 10 h light, 21°C, 60% humidity. Two weeks after germination, seedlings were carefully transferred to individual pots. For in vitro work, Arabidopsis seeds were surface-sterilized by chlorine gas exposure for 4 hours in a sealed desiccator. Seedlings were grown in ½ Murashige and Skoog medium, with 1% sucrose, pH adjusted with KOH 1 M at 5.80 ± 0.02 and 0.5% Phytagel. The chromatin remodeling ATPase mutants were purchased from the European Arabidopsis Stock Centre and are listed in Supporting Information S4 Table. Primers used to genotype the chromatin remodeling ATPase mutants are listed in S4 Table Chromatin remodeling ATPase T-DNA insertion mutants and primers for genotyping (primers obtained from with T-DNA primer design, http://signal.salk.edu/tdnaprimers.2.html). For Table 1, chromatin remodeling ATPases were sorted by protein phylogeny phylogeny.fr [61].

### Bacterial infection assays

*Pst* DC3000 strains were grown overnight in liquid King's Broth (KB) to obtain an OD600 of 1.0. A bacterial suspension of OD600 = 0.1 (equivalent to $5 \times 10^7$ colony-forming units/mL) was prepared in 10 mM MgCl2, 0.04% Silwet L-77 (Lehle Seeds) for spray inoculation. An OD600 = 0.001 bacterial suspension (equivalent to $5 \times 10^5$ colony-forming units/mL) was prepared in 10 mM MgCl2 for syringe-infiltration inoculations. Six 5-week-old plants per genotype were inoculated. Before spray inoculation, plants were labelled and randomly reallocated intermixing lines to avoid position bias. Spray inoculation was performed with a Sparmax TC-620X spray paintbrush (The AirbrushCompany, UK) at a pressure of 1 bar until the whole leaf

surface was completely wet. Infected plants were kept in high humidity for 0 to 3 days. 0.5 cm2 leaf discs were collected with a disc borer. Two leaf discs were collected per plant. Leaf discs from the same line and treatment were combined in pairs, avoiding pairing discs from the same plant and avoiding repeating the same pair combinations. The tissue was ground in 2 mL tubes containing two metallic beads (3 mm diameter) and 200 μL 10 mM MgCl2, with two pulses of 28 Hz for 30 seconds. The suspension was serially diluted with 10 mM MgCl2 and serial dilutions were plated on KB-agar containing the required antibiotics. Bacterial colonies were counted 24 h later, then means and standard deviations were calculated and Two-tailed Student T-test performed, assuming equal variance. Each experiment was repeated independently 3 times.

## Ion leakage experiment

*Pst* DC3000 strains were cultured in the same way as above described. An OD600 = 0.1 bacterial suspension was prepared in 10 mM MgCl2 and syringe-infiltrated into leaves 8 and 9 of 6 different 5-week-old plants. Immediately after infiltration, 0.5 cm$^2$ leaf disks were collected from each infected leaf and incubated in sterile water with for 1 h with mild agitation. The leaf discs were then transferred to 24-well plates containing 1 mL of sterile water placing two discs per well. Every 2 hours, 50 μL of solution were taken to measure conductivity with a conductivity-meter Horiba B-173 Twin Cond (Horiba, Japan).

## Confocal microscopy and FRAP

Samples were prepared from 12 to 15 days *Arabidopsis* seedlings grown in sterile 1/2 MS 1% sucrose or *N. benthamiana* adult leaves. Samples were treated with 100 nM flg22 or mock in liquid medium. After 1 hour, samples were placed with the adaxial surface on the slide glass. Confocal microscopy imaging and fluorescence recovery after photobleaching (FRAP) was performed with a Zeiss LSM 710 (Carl Zeiss Ltd; Cambridge, UK) as previously described [24]. Briefly, an area of 1 μm in radius was bleached in the central section of the nucleus, avoiding the nucleolus, with a three-channel laser (458, 488 and 514 nm) 100% power, and 18 iterations. Subsequently, the nucleus was imaged every minute for 30 minutes. FRAP recovery curves were generated from raw images processed with ImageJ software (https://imagej.nih.gov/ij/). Relative recovery was normalized to total nucleus intensity and background noise, according to Rosa et al., [24].

## Total glutathione analysis

Total leaf glutathione (GSH) was determined spectrophotometrically as the rate of sulfhydryl reagent 5,5′-dithio-bis(2-nitrobenzoic acid) (DTNB) reduction by GSH in the presence of the recycling couple yeast glutathione reductase (GR) and NADPH (Sigma) as described by Rahman *et al.*, [62]. Briefly, leaf tissue was ground in liquid nitrogen and homogenized in 0.1 M potassium phosphate buffer, pH 7.5, 1 mM EDTA, 0.1% Triton x-100 and 25 μM sulphosalicylic acid (1:10 m/v). After centrifugation at 3,000xg for 4 min at 4˚C, the supernatant was incubated in 0.1 M potassium phosphate buffer, pH 7.5, 1 mM EDTA, 0.6 mM DTNB, 0.25 mM NADPH, 1 UN/mL GR. Absorbance was measured in a Tecan Infinite M200 Pro Plate reader (Tecan Trading AG, Switzerland) at 412 mm in intervals of 35 seconds. Values were compared against a standard curve determined with reduced L-GSH.

## RNA extraction and qPCR

Plant tissue for RNA extraction was frozen in liquid nitrogen after harvesting and ground with a pre-chilled drill borer fitting a 2 mL micro-centrifuge tube. Immediately, 1 mL of TRIzol

Reagent (Thermo Fisher Scientific) was added for RNA extraction following manufacturer's instructions. RNA samples were treated with TURBO DNase (AM1907, Ambion, Thermo Fisher Scientific) following manufacturer's instructions. RNA quality was assessed on a 1% agarose gel electrophoresis, and the concentration and purity were measured with a spectrophotometer NanoDrop ND-1000 (Thermo Fisher Scientific). 2 μg of RNA were reverse-transcribed with SuperScript II (18064, Thermo Fisher Scientific), following manufacturer's instructions, using a primer for polyA tails. Quantitative PCR (qPCR) was performed with SYBR Green JumpStart Taq ReadyMix (S4438, Sigma), following manufacturer's recommendations (primers used for qPCR are listed in S5 Table). Three technical replicates were used for each sample. A 384-well plate CFX384 Touch Real-Time PCR Detection System (Bio-Rad Laboratories) and a 96-well plate Mx3005P qPCR System (Agilent Technologies) were used and data was analyzed with the $\Delta\Delta C_T$ method. The average of three genes with constant expression levels at the studied conditions were used as reference for the total messenger RNA concentration: *ACTIN 8* (*ACT8*), *alpha-TUBULIN* (*α-TUB*) and *TIP41*-like family gene (*TIP41*) (S5 Table). All qPCR primers were tested for 90–105% efficiency on a standard curve with 6 template concentrations (10-fold diluted from 0.01 ng/μL for the highest concentration).

## ChIP-qPCR assay

ChIP-qPCR experiments were used to determine the possible association of EDA16 to 4 selected potential targets (*PRX52*, *HSFA2*, *HSP17.6A* and *NATA1*). ChIP-qPCR assays were performed on leaves from 5-week-old Col-0 (control) and Col-0 *35S::EDA16-YFP* plants. For the generation of Col (35S::EDA16-YFP) plants EDA16 was amplified from Col0 cDNA using primers EDA16_cDNA_F1 and EDA16_cDNA_R1 (S5 Table) and cloned into pEG101. The resulting construct was introduced in *Agrobacterium tumefaciens* GV3101 which was used for floral dipping of Col-0 plants. Transformed seeds were selected on glufosinate-ammonium (20 mg/L). Furthermore, the same construct was used for *Agrobacterium*-mediated transient transformation of *N. benthamiana* leaves to confirm expression and nuclear localization of the fusion protein by confocal microscopy using a Zeiss LSM 710 (Carl Zeiss Ltd; Cambridge, UK) (S5A Fig).

The protocol described by Kim et al. [63] was used for chromatin-protein complexes isolation from Col0 and Col0 (*35S::EDA16-YFP*) with minor modifications: initial crosslinking was performed with 1% formaldehyde by vacuum infiltration prior to freezing; a Bioruptor sonicator (Diagenode) was used to break the chromatin into fragments smaller than 500 bp; GFP-Trap Agarose (Chromotek gta-20) were used for immunoprecipitations; the resulting DNA was purified with a QIAquick PCR purification kit (Qiagen) following the manufacturer's instructions. The final samples were used for qPCR using SYBR Green JumpStart Taq ReadyMix (S4438, Sigma) according to the protocol described in the previous section. The relative quantification was performed following the $\Delta C_T$ method, and the input values were used to normalize and calculate the ‰ of input. To increase chances of finding the potential association sites, 3 different primer pairs (localized at the promoter, around the transcription start site, TSS, and in the gene body, respectively) were designed for each of the genes (S5 Table).

## Plant treatment and preparation for RNA-seq and MNase-seq

For the sequencing experiments, two independent biological replicates were prepared and processed independently. For each replicate, ~200 Arabidopsis seedlings were grown on ½ MS solid medium with a long photoperiod of 16 h light, 21°C. After 2 weeks, seedlings were transferred to ½ MS liquid medium overnight in two beakers sealed with Micropore Medical Tape.

The next day, the liquid medium was removed and samples were treated with 100 nM flg22 in ½ MS liquid or ½ MS liquid (mock) for 2 h. Then, samples were removed from the liquid media, dried on paper towel and frozen in liquid nitrogen. Frozen tissue was thoroughly ground to fine powder in liquid nitrogen using a pre-chilled pestle and mortar.

### RNA extraction for RNA-seq

RNA was extracted with the NucleoSpin RNA kit (Macherey-Nagel) starting from ~100 mg of powder, following manufacturer's instructions. RNA purity was assessed by Nanodrop and accurate concentrations were measured with a Qubit RNA HS Assay Kit. RNA library prep was carried out with a #E7420 S/L NEBNext Ultra Directional RNA Library Prep Kit for Illumina (New England Biolabs) following manufacturer's instructions. Agencourt AMPure XP Beads (#A63881, Beckman Coulter, Inc.) magnetic beads were used for RNA purification. RNA libraries were assessed for size quality with a Bioanalyzer, and single-end sequenced with the NextSeq 550 Illumina sequencer.

### RNA-seq data analysis

After quality controls of raw sequencing data with FastQC, untrimmed data sequences were mapped with STAR [64] to the Arabidopsis TAIR10 genome, followed by read counting with HTseq-count implemented with LiBiNorm [65,66], using the following parameters:— order = pos—minaqual = 10—mode = intersection-nonempty—idattr = gene_id—type = exon —stranded = reverse. The data counts were normalised and analyzed with the R package DEseq2 [67]. To compare the flg22-treated and mock-treated samples a model accounting for the treatment and the genotype excluding a replicate effect was used: "~condition + replicate". To establish the differences caused by the flagellin treatment flg22-treated versus mock-treated samples were compared pairwise (Col-0_mock vs. Col-0_flg22 and so on for the mutants). Finally, mutants and the distinct effect of the treatment on the mutants were addressed by comparing pairwise between them (Col-0 and *eda16-OE*, Col-0 and *eda16-ΔHc* and *eda16-OE* and *eda16-ΔHc*) both for the mock-treated and flg22-treated samples and filtering for flg22-regulated genes in Col-0. The adjusted p-values accepted for significance were $< 0.05$ with a fold-change $> 1.5$.

### Nuclei extraction, MNase digestion and library preparation

Frozen powder (2 g) was used for nuclei extraction with 10 mL of nuclei extraction buffer 1 (0.4 M sucrose, 10 mM Tris/HCl, pH 8.00, 10 mM $MgCl_2$, 5 mM ß-mercapto-EtOH, 0.1 mM PMSF and Protease Inhibitor Mix P, 39103 Serva), filtering debris out through a 200 µm filter and centrifuging supernatant at 1000 g for 10 min at 4˚C. Nuclei pellet was washed in 5ml of nuclei extraction buffer 2 (25 mM Sucrose, 10 mM Tris/HCl pH 8.00, 10 mM $MgCl_2$, 1% Triton X-100, 5 mM ß-mercapto-EtOH, 0.1 mM PMSF and Protease Inhibitor Mix P, 39103 Serva), mixed by vortex, filtered using a using a 60 µm filter and centrifuged at 1000 g for 10 min at 4˚C. The nuclei pellet was rinsed with microccocal nuclease (MNase) buffer (10 mM Tris-HCl pH 7.5, 15 mM NaCl, 60 mM KCl, 1 mM $CaCl_2$, 0.15 mM Spermine and 0.5 mM Spermidine) and re-suspended in 250 µL of MNase buffer. The DNA concentration was quantified with a NanoDrop, and samples were diluted to 400 ng/µL. 1 µl of 25 U/µL microccocal nuclease (MNase) was added to 125 µL per sample and incubated at 37˚C for 10 minutes. To stop the reaction, 125 µL of Stop Buffer 2x (50 mM EDTA, 50 mM EGTA and 1% SDS) Tris/HCl pH 6.50 and 4 µL of proteinase K (stock 10 mg/ml) were added and incubated at 45˚C for 1h. Samples were purified with a QIAquick PCR Purification Kit (Qiagen) and eluted with 15 µL water. The eluate was loaded onto a 1% agarose gel (without loading buffer dye) and the

lowest band (mono-nucleosome DNA) was excised and gel-purified with QIAquick Gel Extraction Kit (Qiagen). Libraries were prepared starting from 50 ng of DNA per sample. DNA library prep was carried out with a NEBNext Ultra II DNA Library Prep Kit for Illumina (E7645S/L, NEB) following manufacturer's instructions. Agencourt AMPure XP Beads (#A63881, Beckman Coulter, Inc.) magnetic beads were used for DNA purification. DNA libraries were assessed for size quality with a Bioanalyzer and single-end sequenced with the NextSeq 550 Illumina sequencer (GEO Series accession number GSE149654).

### MNase-seq analysis

Raw reads were trimmed with Trimmomatic (see parameters: SE -threads 8 -phred33), mapped with bowtie2 (-p 8—very-sensitive -x) to the Arabidopsis TAIR10 genome. Sorted Bed files were analyzed with Danpos2 using function dpos with FDR < 0.01 in order to call as a Differentially Positioned Nucleosome (PDN). Small in-house scripts were written in C++ in order to produce the phasograms [68] and to map DPNs nucleosomes to genes including 1000 base pairs upstream of the TSS as promoter region. K-means clustering was performed in R with "kmeans" package. The RNA-seq and MNase-seq data from this publication have been deposited to the NCBI's Gene Expression Omnibus (http://www.ncbi.nlm.nih.gov/geo/) and are accessible through GEO Series accession number GSE149654.

## Supporting information

**S1 Fig. flg22 treatment promotes histone nuclear diffusion and affects gene expression and nucleosome positioning at specific loci but does not alter nucleosome phasing.** (A) FRAP data collected from seedling leaf tissue H2B-GFP in Col-0 or (B) transient expression in *Nicotiana benthamiana* adult leaves. The tissue was exposed to water or 100 nM flg22 for 1 hour before imaging. Data points are averages of at least 8 nuclei for each condition. Error bars represent standard error of the mean. (C) flg22-regulated genes. RNA-seq gene expression scatter plot showing Differentially Expressed Genes (DEGs, adjusted p-value < 0.05, fold-change > 1.5) on 2-week-old *Arabidopsis* seedlings (Col-0) following elicitation with 100 nM flg22 compared with mock; induced (yellow), unaltered (grey) and repressed genes (blue). (D) flg22 elicitation does not change the average genomic nucleosome phasing. Nucleosome phasogram of Col-0 plants following 100 nM flg22 treatment (red) and control (black). On top right corner linear correlation fit between nucleosome peak and base pairs (bp). Red, treatment (slope = 177.37 bp/nucleosome) and black control (slope = 177.37 bp/nucleosome). (E) Nucleosome fuzziness. Analysis of nucleosome fuzziness at mock state (x-axis) compared with 100 nM flg22 treatment (y-axis) using Dynamic Analysis of Nucleosome Position and Occupancy by Sequencing (DANPOS, FDR < 0.01). (F) Nucleosome summit intensity. Analysis of nucleosome peak at mock state (x-axis) compared with 100 nM flg22 treatment (y-axis) using DANPOS (FDR < 0.01).
(TIF)

**S2 Fig. Early MTI responses and ETI are not affected by the EDA16 mutation.** (A) Accumulation of *FRK1* (left), *PHI-1* (middle) and *CBP60g* (right) transcripts was assessed by qPCR in 2-week-old Col-0 (black) *eda16-OE* (blue), *eda16* line SALK_208691 (grey) and *eda16-ΔHc* (red) seedlings elicited with 100 nM flg22. Values are average of three biological repeats ± SE presented as fold induction compared with Col-0 mock-treated sample at time 0. (B) and (C) ETI responses in eda16 mutants. 5-week-old Col-0, *eda16-OE*, *salk_208691*, and *eda16-ΔHc* plants were syringe-infiltrated with *Pst* DC3000 EV or *Pst* DC3000 *avrRpt2*. For Ion leakage leaf disks were collected and kept in sterile water. Conductivity measurements (microsiemens

per meter) were taken from the solution at different times as indicated (B). Bacterial colony forming units were determined 3 days post-infection (C). Error bars represent standard deviation (n = 6) and the experiment has been repeated 3 times with identical results. Differences were not statistically significant (two-sided T-test) between Col-0 and the *eda16* mutants.
(TIF)

**S3 Fig. flg22-dependent gene expression and nucleosome phase changes in the *eda16* mutants.** flg22-dependent gene expression and nucleosome phase changes in the *eda16* mutants. (A) flg22 elicitation does not change the average genomic nucleosome distribution in the eda16 mutants. Nucleosome phasogram of Col-0, *eda16-OE* and *eda16-ΔHc* plants before (mock) and after elicitation with flg22 (100 nM). (B) EDA16 affects flg22-regulated genes. Principal component analysis (PCA) of RNA-seq normalized read count data reveals a greater difference in gene expression between flg22- and mock-treated plants (principal components 1, PC1 and PC2, accounting between the two for near ~80% of the variance) than between different genotypes (clustered by PC3 and PC4, accounting between the two for ~13% of the variance). (C) Gene count distributions in Col-0 and eda16 mutants following elicitation with flg22. Normalized count distributions are displayed as boxplots for Col-0, *eda16-OE* and *eda16-ΔHc* for mock-treated or elicited with flg22 (100 nM) plants.
(TIF)

**S4 Fig. The *EDA16* mutation alters glutathione levels.** (A) The *eda16-ΔHc* mutant has elevated glutathione (GSH). Basal total glutathione (GSH) levels were determined in 3-week-old Col-0, *eda16-OE* and *eda16-ΔHc* plants. Error bars represent standard deviation, n = 3. Statistical differences are indicated (two-sided T-test p-values: * < 0.05). (B) *EDA16* regulates the expression genes involved in glutathione production. Gene expression heatmap for genes involved in glutathione production between Col-0, *eda16-OE* and *eda16-ΔHc* plants 2h after elicitation with 100 nM flg22.
(TIF)

**S5 Fig. EDA16 promotes nucleosome remodeling at redox-related loci.** (A)EDA16 is localized in the nucleus. Confocal localization of 35S::EDA16-YFP construct. (B) EDA16-mediated nucleosome repositioning. IGV image of MNase-seq reads over *HSFA*, *HSP17.6A*, *NATA1*, and *PRX52* loci for Col-0 (top), *eda16-OE* (middle), and *eda16-ΔHc* (bottom) as indicated. Tracks for mock (grey) and flg22 (pink) conditions are overlaid. Primers used in Fig 5E indicated below their respective IGV gene track in orange.
(TIF)

**S1 Table. Sequencing mapping statistics for MNnase-seq (Supports Figs 1–5).**
(DOCX)

**S2 Table. Differentially positioned nucleosomes (DPNs) between flg22- and mock treated samples (Col-0, *eda16-OE* and *eda16-ΔHc*) detected with any of DANPOS parameters, summit, point or nucleosome fuzziness.**
(DOCX)

**S3 Table. Differentially positioned nucleosomes (DPNs) between flg22- and mock treated samples mapped to protein-coding gene regions (promoters; including -1000 bp from TSS, TSS; including 180 bp, +/- 90 bp from TSS, gene bodies; from TSS to TES and promoters + gene bodies).**
(DOCX)

**S4 Table. Chromatin remodelling ATPase T-DNA insertion mutants and primers for genotyping and cloning *EDA16* cDNA.**
(DOCX)

**S5 Table. Primers for qPCR and ChIP-qPCR.**
(DOCX)

**S1 Dataset. Col-0 flg22 Differentially expressed genes.**
(XLSX)

**S2 Dataset. Col-0 flg22-altered nucleosomes mapped to genes.**
(XLSX)

**S3 Dataset. GO term analysis of flg22-induced genes with DPNs.**
(XLSX)

**S4 Dataset. GO term analysis of non-DEGs with DPNs.**
(XLSX)

**S5 Dataset. GO term analysis of flg22-repressed genes with DPNs.**
(XLSX)

**S6 Dataset. Differential nucleosome deposition K-means clustering of flg22-induced genes with DPNs.**
(XLSX)

**S7 Dataset. Differentially positioned nucleosomes (DPNs) between flg22- and mock treated samples.**
(XLSX)

**S8 Dataset. *eda16-OE* DEGs mock vs flg22.**
(XLSX)

**S9 Dataset. *eda16-OE* flg22-mediated DPNs mapped to genes.**
(XLSX)

**S10 Dataset. *eda16-ΔHc* DEGs mock vs flg22.**
(XLSX)

**S11 Dataset. *eda16-ΔHc* flg22-mediated DPNs mapped to genes.**
(XLSX)

**S12 Dataset. GO term analysis of flg22-dependent EDA16-regulated genes with DPNs.**
(XLSX)

**S13 Dataset. flg22-differentially expressed genes in the 3 genotypes.**
(XLSX)

## Acknowledgments

We thank Dr Miriam L. Gifford, Daniela J. Sueldo, Stephanie Kancy and Ruth Eichmann for critically reading the manuscript, all members of the Ntoukakis' laboratory for fruitful discussions and helpful comments.

## Author Contributions

**Conceptualization:** Alonso J. Pardal, Steven H. Spoel, Moussa Benhamed, Vardis Ntoukakis.

**Data curation:** Alonso J. Pardal, Emma Reilly, David Latrasse, Lorenzo Concia.

**Formal analysis:** Alonso J. Pardal, Sophie J. M. Piquerez, Ana Dominguez-Ferreras, Lucas Frungillo, Emmanouil Mastorakis, Emma Reilly, David Latrasse, Lorenzo Concia, Selena Gimenez-Ibanez.

**Funding acquisition:** Alonso J. Pardal, Lucas Frungillo, Steven H. Spoel, Vardis Ntoukakis.

**Investigation:** Alonso J. Pardal, Sophie J. M. Piquerez, Ana Dominguez-Ferreras, Lucas Frungillo, Emmanouil Mastorakis, David Latrasse, Lorenzo Concia, Selena Gimenez-Ibanez, Vardis Ntoukakis.

**Methodology:** Alonso J. Pardal, Vardis Ntoukakis.

**Project administration:** Vardis Ntoukakis.

**Supervision:** Steven H. Spoel, Moussa Benhamed, Vardis Ntoukakis.

**Writing – original draft:** Alonso J. Pardal, Ana Dominguez-Ferreras, Moussa Benhamed, Vardis Ntoukakis.

**Writing – review & editing:** Alonso J. Pardal, Sophie J. M. Piquerez, Ana Dominguez-Ferreras, Lucas Frungillo, Emmanouil Mastorakis, Emma Reilly, David Latrasse, Lorenzo Concia, Selena Gimenez-Ibanez, Steven H. Spoel, Moussa Benhamed, Vardis Ntoukakis.

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
