## [Decision Letter · Decision Letter 0]

16 Nov 2020

Dear Dr Ntoukakis,

Thank you very much for submitting your manuscript "Immunity onset alters plant chromatin and utilizes EDA16 to regulate oxidative homeostasis" for consideration at PLOS Pathogens. As with all papers reviewed by the journal, your manuscript was reviewed by members of the editorial board and by several independent reviewers. The reviewers are overall enthusiastic, but raise some significant (and largely overlapping) concerns. In light of the reviews (below this email), we would like to invite the resubmission of a significantly revised version that takes into account the reviewers' comments.

We cannot make any decision about publication until we have seen the revised manuscript and your response to the reviewers' comments. Your revised manuscript is also likely to be sent to reviewers for further evaluation.

Sincerely,

David Mackey

Associate Editor

PLOS Pathogens

Bart Thomma

Section Editor

PLOS Pathogens

Kasturi Haldar

Editor-in-Chief

PLOS Pathogens

orcid.org/0000-0001-5065-158X

Michael Malim

Editor-in-Chief

PLOS Pathogens

orcid.org/0000-0002-7699-2064

Reviewer's Responses to Questions

**Part I - Summary**

Reviewer #1: In this manuscript Pardal et al. report the function of the chromatin remodeler EDA16 in the transcriptional regulation of defense responses against bacterial infection. The authors characterize changes in nucleosome occupancy after flg22 infection and correlate this with altered gene expression. From a reverse genetics approach they identify eda16 as a mutant that is more resistant to infection with Pseudomonas syringae than the wild type. Interestingly, an overexpressing line is more susceptible. The authors attribute this phenotype to altered nucleosome occupancy at a group of genes that have been linked to redox homeostasis. To lend support to this hypothesis, they determine the redox status by determining GSH levels. Overall, the manuscript is interesting and well written and the experiments appear to be well conducted. I have, however, a number concerns regarding the interpretation of the findings as detailed below.

Major comments:

- The title states that “EDA16 regulates oxidative homeostasis”. The results presented are, however, entirely consistent with an indirect role of EDA16 on oxidative homeostasis, which may or may not be mediated by the group of genes affected in eda16 mutants. Demonstrating direct interaction of EDA16 with the suspected target genes by chromatin immunoprecipitation using an antibody against EDA16 is be an essential experiment to justify such a conclusion.

- The abstract claims that a novel feedback mechanism has been revealed (l. 30). Again, in my opinion this is not supported by the data presented. I fail to see the “feedback” element (the moderate transcriptional upregulation of EDA16?). This statement needs to be clarified (and possibly backed by additional experimentation) or toned down.

- Similarly, the last sentence of the abstract “framework to prevent protracted activation of immunity” is an overstatement. While this is an interesting idea, I wonder whether the effects of eda16 on a group of 10 genes that, as far as I can see, have not previously been recognized to be central to the MTI response, can constitute a general mechanism that would warrant to call it a “framework”. Moreover, while the prevention of protracted gene activation by chromatin remodelling is an interesting hypothesis, the manuscript does not show any timecourse of gene expression that would indicate that genes are active for longer in eds16 mutants and that this causes the enhanced resistance.

- In the Col wild type analysis (Fig. 1) I wonder whether DEGs are enriched in DPNs and if so, whether this is statistically significant, i. e. whether proportionately more genes that are differentially expressed show changes in nucleosome structure than would be expected by chance. It looks a bit like 50% of all genes have DPNs after treatment and 50% of DEGs have DPNs (as rough estimate). This is exactly what you would expect is the two were unrelated effects.

-l. 159: The more relevant control group for this test in enrichment would be a pool of all the clusters except cluster 3. Naively, I would expect a lot of defense-related genes (such as the DEGs) to have WRKY binding sites in their promoters.

Additional comments:

-Several authors are not listed in the Authors Contributions list.

-EDA16 should be spelled out at first mentioning (l. 28 or title)

-l. 54: “relies”

-l. 222: “EXPRESSOR”

Reviewer #2: The study is novel since the role of EDA16 in plant immunity was not known earlier. Authors used MNase-seq and RNAseq to validate their claims. The lack of data on physical association of EDA16 to the DPNs is the major weakness of the manuscript.

Reviewer #3: This MS describes the characterisation of chromatin remodelling factors in Arabidopsis, in response to flg22 treatment. The authors show that stimulation of MTI not only leads to the induction of gene expression (and repression) but also the repositioning of nucleosomes on chromatin, offering a mechanism of MTI-triggered transcriptional reprogramming. Critically, the authors go on to identify factors responsible for nucleosome re-positioning, via a reverse genetics approach.

The authors home in on EDA16, an ATPase domain containing proteins that is part of a family of chromatin remodelling factors. Over expression of EDA16 suppresses immunity, whereas its absence leads to exacerbation of MTI responses. These observations are in line with the finding that EDA16 is induced relatively late during MTI and suggest that EDA16 acts as a negative regulator that helps dampen late MTI responses.

Overall, the work is well presented, carefully considered and the conclusions underpinned by convincing data. Given the importance of MTI in plant immunity and the findings presented, this represents an important body of work that will be of great interest to the plant-microbe interactions community and beyond.

**Part II – Major Issues: Key Experiments Required for Acceptance**

Reviewer #1: - The title states that “EDA16 regulates oxidative homeostasis”. The results presented are, however, entirely consistent with an indirect role of EDA16 on oxidative homeostasis, which may or may not be mediated by the group of genes affected in eda16 mutants. Demonstrating direct interaction of EDA16 with the suspected target genes by chromatin immunoprecipitation using an antibody against EDA16 is be an essential experiment to justify such a conclusion.

- The abstract claims that a novel feedback mechanism has been revealed (l. 30). Again, in my opinion this is not supported by the data presented. I fail to see the “feedback” element (the moderate transcriptional upregulation of EDA16?). This statement needs to be clarified (and possibly backed by additional experimentation) or toned down.

- Similarly, the last sentence of the abstract “framework to prevent protracted activation of immunity” is an overstatement. While this is an interesting idea, I wonder whether the effects of eda16 on a group of 10 genes that, as far as I can see, have not previously been recognized to be central to the MTI response, can constitute a general mechanism that would warrant to call it a “framework”. Moreover, while the prevention of protracted gene activation by chromatin remodelling is an interesting hypothesis, the manuscript does not show any timecourse of gene expression that would indicate that genes are active for longer in eds16 mutants and that this causes the enhanced resistance.

Reviewer #2: Major Comments:

1. In the present manuscript, authors have shown that EDA16 (member of Ris1 subfamily) is involved in nucleosome repositioning in plant defense. MNase-seq and RNA-seq were performed in Col-0 and 2hrs flg22 treated Col-0. The similar experiments were also performed in eda16-OE and eda16-∆Hc to identify DPNs (which could either be enriched nucleosomes or depleted showing either increase in occupancy or decrease) over promoter or genic region (Line 254-268, 139-144), the contrasting DPNs were also observed in eda16-OE and eda16-∆Hc. Thus, authors concluded the involvement of EDA16 in chromatin remodeling during plant defense. Considering EDA16 is ATPase chromatin remodeler, it would be nice if they clearly classify DPNs into enriched and depleted and try to correlate their RNAseq data with those enriched/depleted nucleosomes. It will help in clearly depicting the role EDA16 in chromatin remodeling.

2. To ascertain the role of EDA16 in remodeling of DPNs, they should also perform ChIP to confirm physical association of EDA16 around DPNs.

Reviewer #3: The authors infer a model in which EDA16 dampens the expression of flg22 induce genes at the later timepoints. This is based on the observation that EDA16 is expressed relatively late after elicitation. If this is the case, it would make sense for the authors to test a number of flg22 and EDA16 responsive genes in a time course, assessing whether their expression drops in an EDA16-dependent manner. Testing a representative set of genes would be enough for that.

**Part III – Minor Issues: Editorial and Data Presentation Modifications**

Reviewer #1: - In the Col wild type analysis (Fig. 1) I wonder whether DEGs are enriched in DPNs and if so, whether this is statistically significant, i. e. whether proportionately more genes that are differentially expressed show changes in nucleosome structure than would be expected by chance. It looks a bit like 50% of all genes have DPNs after treatment and 50% of DEGs have DPNs (as rough estimate). This is exactly what you would expect is the two were unrelated effects.

-l. 159: The more relevant control group for this test in enrichment would be a pool of all the clusters except cluster 3. Naively, I would expect a lot of defense-related genes (such as the DEGs) to have WRKY binding sites in their promoters.

Additional comments:

-Several authors are not listed in the Authors Contributions list.

-EDA16 should be spelled out at first mentioning (l. 28 or title)

-l. 54: “relies”

-l. 222: “EXPRESSOR”

Reviewer #2: Minor Comments:

1. In line 114, author should also provide the pvalue (statistical significance) for the length (177 bp) between the nucleosomes peaks.

2. Author has performed MNase-seq in order to identify the nucleosomes positioning. The authors should provide the table for MNase-seq data regarding number of million paired-end read generated and coverage on the Arabidopsis genome.

3. In line 124, the three parameters compared by DANPOS: nucleosomes fuzziness, summit intensity or point location. What they are representing should be provide either in introduction or in results.

4. In Fig2A, authors have performed k-mean clustering to classify the 1142 flg22-induced genes and then identify motifs enriched in cluster3. The similar type of analysis should also be done on the 242 flg22-repressed gene.

5. The authors have chosen 20 chromatin remodeling ATPase mutants and then spray inoculated them with a Pseudomonas syringae pathovar tomato DC3000 (Pst DC3000). Further, they have chosen EDA16 on the basis of FC since in EDA16, the FC was highest and showed resistance phenotype. Since, with the advancement in the plant age there is tremendous change in the transcriptome profile of the plants. The author should provide the logical reason for the comparison since, the FC identified from RNA-seq data at 2 week old plants while the experiment was performed at 5- week old plants.

6. In fig 3B, authors have shown that expression of EDA16 is 3 hours following infection with Pst DC3000. It will be appropriate, if authors also provide the qPCR data of the intermediate time points also.

7. In line 290-300, authors have identified few genes (21genes) having change in the expression levels between the two mutants as well as with the Col-0. With that, these genes also have distinct nucleosome dynamics. All the performed analysis is the result obtained from the data generated from the sequencing platform. The authors should also provide experimental validation from the same.

8. In fig 3H, authors have shown the log10(cfu/cm2 ). The result obtained showed the comparison when inoculated with DC. The authors should cross-check once their data as their error bars are almost overlapping while the pvalue obtained is “ ** < 0.01 “.

9. Author should perform following small modification:

a. Use “*” where necessary in the figure to show the statistical significance of the value like in fig 3A, 3E.

b. Similarly, in entire test, please use the similar format for a particular word. Like in some places pval is written while at other place pvalue is written.

c. Line 127, capitalize the legend number of h,i,j.

Reviewer #3: To improve the MS, I would like for the authors to consider the following points:

- It would be useful for the authors to state how similar their flg22 transcriptome datasets agrees with the literature. Is there significant overlap between studies or differences as well?

- The authors look at nucleosome occupancy, responsible for DEG and directed by EDA16. It would be of interest to see to what extent, other modelling factors are under the control or are affected by MTI and EDA16. Is there cross-talk between regulators? Are there distinct ‘mmune sectors’ that are controlled by EDA16?

- Now that the role of EDA16 has been established and it has emerged as an important regulator, one critical question is: how is EDA16 expression regulated? It is clear that WRKY TFs play important roles in MTI, so the obvious question is as to whether cognate responsive elements can be found in the Eda16 promoter. Similarly, and to identify possible feedback loops, it would be interesting to know whether EDA16 itself could be a target and experiences DNP. Analyses of the promoter regions in the OE and mutant lines (for which data is available) may offer yet more interesting clues.

- In the MS, a number of datapoints, that could prove to be revealing, are not described. For example, in Figure 4, a set of genes appear to be positively regulated by EDA16 as these genes are highly induced during OE (in both mock and flg22 treated plants) and repressed in the mutant. What kind of proteins do they encode?

- A weaker aspect of this MS relates to the changes in redox state, attributed to EDA16 action. While the data, demonstrating changes, is convincing, the induction of GST and changes in redox status could be due to downstream effects (induction/repression of genes controlled by EDA16). While I appreciate the difficultly untangling cause and effect relationships, the authors should be careful to not over-interpret their results in that regard.

- The observation that ETI is not affected by EDA16 is interesting and given its established role in MTI regulation, not surprising. In that context, I would encourage the authors to speculate as to why this could be. One hypothesis that could be put forward is that ETI often features (and requires?) drastic responses to pathogen ingress, eliminating the need for the dampening of responses. Alternatively, other modelling factors could be at play.

PLOS authors have the option to publish the peer review history of their article (what does this mean?). If published, this will include your full peer review and any attached files.

Reviewer #1: No

Reviewer #2: No

Reviewer #3: No
---

## [Editor Report · Decision Letter 1]

19 Apr 2021

Dear Dr Ntoukakis,

We are pleased to inform you that your manuscript 'Immunity onset alters plant chromatin and utilizes EDA16 to regulate oxidative homeostasis' has been provisionally accepted for publication in PLOS Pathogens.

Best regards,

David Mackey

Associate Editor

PLOS Pathogens

Bart Thomma

Section Editor

PLOS Pathogens

Kasturi Haldar

Editor-in-Chief

PLOS Pathogens

orcid.org/0000-0001-5065-158X

Michael Malim

Editor-in-Chief

PLOS Pathogens

orcid.org/0000-0002-7699-2064
---

## [Editor Report · Acceptance letter]

14 May 2021

Dear Dr Ntoukakis,

We are delighted to inform you that your manuscript, "Immunity onset alters plant chromatin and utilizes EDA16 to regulate oxidative homeostasis," has been formally accepted for publication in PLOS Pathogens.

Best regards,

Kasturi Haldar

Editor-in-Chief

PLOS Pathogens

orcid.org/0000-0001-5065-158X

Michael Malim

Editor-in-Chief

PLOS Pathogens

orcid.org/0000-0002-7699-2064